# Ultra-Low-Level Laser Therapy and Acupuncture Libralux: What Is so Special?

**DOI:** 10.3390/medicines6010040

**Published:** 2019-03-14

**Authors:** Luca Evangelista, Bruno De Meo, Gianluca Bernabei, Gabriele Belloni, Giovanni D’Angelo, Marzio Vanzini, Laura Calzà, Michele Gallamini

**Affiliations:** 1MMDD Centre of Pain Therapy and Angiology Cassino (FR), 03043 Cassino (FR), Italy; Luca.evangelista1@libero.it (L.E.); bruno.demeo@yahoo.it (B.D.M.); 2FKT Fisiocrea Srl Baldissero Canavese, 10080 Baldissero Canavese (TO), Italy; gianluca@damanhur.it; 3FKT Freelance, Via C. Colombo, 00124 Rome, Italy; gabriele26@gmail.com; 4FKT Rehability Center Padova, 35132 Padova, Italy; fisiodangelo@gmail.com; 5MD Ophthalmologist Acupuncturist Oculistica Viva Eye Clinic Bologna, Via U. Lenzi, 40122 Bologna, Italy; marzvan@tin.it; 6MD Prof. IRET Foundation, 40064 Ozzano Emilia, Bologna, Italy; Interdepartmental Centre for Industrial Research in Health Sciences and Technologies, University of Bologna, 40136 Bologna, Italy; laura.calza@unibo.it; 7Eng. Freelance MD Consultant, Sal. Maggiolo di Nervi, 16167 Genova, Italy

**Keywords:** laserpuncture, musculoskeletal disorder, musculoskeletal pain, photobiomodulation, pulse-modulated laser emission

## Abstract

**Background**: Contrary to the most credited theories on laser therapy that see power/energy as the major factors to its effectiveness, a technique using an extremely low power/energy laser stimulation to treat musculoskeletal pain and dysfunction is proposed. The stimulus consists of a 20 s train of modulated pulses with an average power below 0.02 mW and is applied on sequences of acupuncture points selected according to the impaired segment of the patient’s body. **Methods**: Modifications on the extracellular soft tissue matrix and on the “fascia” were sonographically demonstrated. Laboratory and clinical tests confirmed the effectiveness. **Results**: Responses similar to those experienced in acupuncture were observed. The device—a CE Class IIa certified medical device named Libralux—affords a clinically proven effectiveness exceeding 80% in the treatment of musculoskeletal conditions and associated motor dysfunctions. An average of just three application sessions was generally sufficient to overcome the dysfunction. **Conclusions**: The development of the method is supported by over 20 years of R&D activities, with a range of experiments discussed in several papers published in indexed peer-reviewed journals. A few considerations regarding the possible physiological action mechanisms involved are proposed in this paper.

## 1. Introduction

Current knowledge [1] regarding medical lasers is converging on the idea that power and energy doses are crucial to achieve the specific healing effect [2]. This idea is correct insofar as the laser radiation, concentrated in a coherent beam capable of penetrating the tissues, can promote a biochemical reaction within the cells of the injured or affected organ. For this reason, the laser application is local, i.e., if your right shoulder is affected, you receive a laser treatment on your right shoulder, exactly on the affected structures.

A certain level of power/energy is required to obtain such effects, but over a given limit, the energy can be destructive or have negative effects, while under a lower limit, it has been demonstrated that no effects can be achieved. The Arndt and Schultz Law [2] stating the principle is shown in Figure 1.

Typical power/energy doses are recommended for these applications by the World Laser Association [3].

The Libralux output is at a much lower power/energy, well under any known limit [4] (see the Libralux energy plotted on the graph of Figure 1), because it was created on different criteria that will be detailed in the following paragraph.

Despite such a tiny stimulation energy, when applied on distal acupuncture points (APs), there is rather sound evidence of its effectiveness.

The purpose of this paper is to provide a review of the available evidence to suggest a possible physiological action mechanism.

## 2. Libralux Main Characteristics

Libralux has been designed on the experiences previously collected with Biolite^®^ radiating a 0.03 mW average power with a 100 Hz, 1% Duty Cycle red light laser emission. The basic idea was to apply a level of energy similar to the one of the solar radiations that accompanied the life evolution. The second one was to pulsate that energy at a frequency in the range of the known physiologic oscillations. The intended effect was an interaction with normal physiology in the extracellular soft tissue matrix rather than producing deep effects on defective tissues.

Through a computer-assisted interface (Figure 2), Libralux provides trains of 0.1 ms low-power red laser pulses according to a combination of ON/OFF modulations, including:(a)A meridian resonance frequency (12 different values in the band 5–11 Hz);(b)An anti-addiction 1 Hz frequency.

The train is automatically set at a 20 s duration.

Depending on operator choice and the specific dysfunction, the appropriate sequence of APs is suggested with a visual guide.

The laser emission has the following characteristics:Laser wavelength (nm)     650 (red light)Peak emitted power (mW)     7Beam divergence (mrad)    35Spot size @ 20 mm (cm^2^)     0.2Spot size @200 mm (cm^2^)    20Modulated Emission:
◦Carrier Frequency       100 Hz Duty Cycle 1%◦Meridian Modulation    5–11 Hz Duty Cycle 50%◦Antiaddiction Modulation  1 Hz Duty Cycle 50%

Mean Stimulation Power     0.0175 mWMean Stimulation Energy (20 s)  0.35 mJ

The anti-addiction frequency (1 Hz) is included to prevent the addiction to enkephalins synthetized by the 100 Hz stimulation, thus increasing the stimulation effects [5].

The meridian resonance frequencies [6] were selected to improve the diffusion of the stimulus along each specific meridian (the lower Schumann resonance band 5.5 to 11 Hz has been selected).

## 3. The Evidence

The device is a development of a previously proposed device, called Biolite^®^, that was the specific subject of a publication [7] describing the main application criteria. After some clinical trials [8,9], where there could have been a placebo effect, several trials on animal models have confirmed the effectiveness of the acupuncture-derived applications in acute, chronic, and neuropathic pain [10,11]. A few in vitro studies demonstrated that in spite of the very low power/energy level, the radiation was capable of inducing significant effects at the cell and tissue level [12,13,14], which was confirmed by tests performed by other scientists [15].

The initial evidence of effectiveness on motor control dysfunctions, especially on balance somatosensory deficit [16], has been confirmed both by multiple case reports [17] and by a pilot study [18].

## 4. The Libralux Application

During the treatment of musculoskeletal pain with Libralux, the following pattern is observed.

The treatment protocols included the stimulation of an average of 8 APs. After the stimulation of each AP, the patient was asked to perform a previously impaired or painful movement. Very frequently, upon the completion of stimulation of the 4th AP, the movement showed remarkable improvement. Our experience then suggested discontinuing the treatment because the improvement would increase in the following eight hours, an experience not uncommon to classical acupuncture.

A further reason for discontinuing the application was that in spite of the tiny energy level applied, there could be an undesirable over-stimulation effect. It was much better to consider a new session—generally at a three-day interval—than to insist on performing the whole protocol at once (Figure 3).

If the pain or dysfunction, in spite of the temporary remission, returned in a few days, the clinician should be aware of an active irritational factor to be found and resolved.

## 5. The Physiological Mechanism

To determine the physiological mechanism, we concentrated on a series of topics.

### 5.1. Musculoskeletal Pain

Musculoskeletal pain (MP) is not always associated with a tissue injury or pathology [19]. More frequently than we perhaps believe, MP can be the effect of a soft tissue modification—generally a contraction of both agonist and antagonist muscles [20]. It can be observed that such modification is accompanied by a new extracellular soft tissue matrix (ECM) status: from the normal SOL state to the GEL condition [21]. The role of GEL and SOL is well explained by Pollack [22]. This modification affects the normal homeostasis of soft tissues, spreading—probably through the neuronal reflexes started by the noxious stimuli—to the surrounding tissue, according to metameric distributions. When in GEL status, a different pH and higher ECM viscosity affect the synaptic connections of the peripheral nerves. The neurotransmitter flow through a pH-modified region is certainly affected and likely to disrupt what, in engineering terms, could be called the automatic gain control of the temporal/spatial summation of noxious stimuli. “Fascia is the soft-tissue component of the connective tissue system that permeates the human body, forming a continuous, whole-body, three-dimensional matrix of structural support. It interpenetrates and surrounds all organs, muscles, bones, and nerve fibers, creating a unique environment for body systems functioning” [23]. Its role is essential to appropriate motor performances [24].

### 5.2. MP and Dysfunction

The modified tissues (co-contracted muscles, Extra Cellular Soft Tissue Matrix (ECM) in GEL status) can both send nociceptive information to the central nervous system and impair regular functions of muscles. This “interference” with proprioceptive information affects control functions both in the feed-back loop of motor control and in the adaptive adjustment of feed-forward motor planning mechanisms [25,26,27]. The impairment is more evident in subjects who, being affected by other pathologies, lack the redundancy of controls that protect a healthy individual (e.g., falls in the elderly [28,29]).

### 5.3. Libralux Effect

It is known that low-energy radiation can promote cellular redox activities [30]. The device, with its pulsating emission, gives the ECM a flow of photons very similar to those that spread through the ECM under normal metabolic conditions [31]. This flow promotes a return to normal metabolic conditions in the ECM, thus restoring the “normal” condition and switching off the nociceptive stimulus.

## 6. The Libralux and Its Effects

A comprehensive discussion was made in a specific paper [4] that was previously cited. A short summary of the evidence presented follows:(a)Normally, ECM proteins have a cyclical oxidation/redox cycle (approximately 100 times/s, similar to the main Libralux modulation) [32] (Figure 4);(b)The average density of ECM proteins [33] has a mean order of magnitude of 10^13^ chains per square centimetre; Libralux affords the same photon density (under the skin, an average power density of 20 nW over a 10 mm^2^ surface = 200 nW/cm^2^);(c)At a body temperature of 37 °C, the oxidation emitted photons have a wavelength of 650 nm, identical to the Libralux emission;(d)While the oxidation process frees two photons, the following redox process requires just one photon; thus, the overflow of photons through the ECM can spread across the body along the ubiquitous ECM, which unlike surrounding tissues is transparent to visible light;(e)To start the process, one must stimulate the synchronization of photon flow from a distance as far away as possible from the affected area [34]; that is why the left leg is treated to obtain an effect on the right shoulder;(f)To access the ECM, Libralux exploits the acupuncture stimulation points (Aps) [35]—a funnel through the dermis and fascia filled with free nerve terminations (which is why APs are so sensitive to pressure) and a significantly higher percentage of small blood vessels [36]. The points have different properties in comparison to the surrounding tissues, including significantly lower electrical impedance [37] and a superior absorbance of visible light [38];(g)The acupuncture meridians are very likely to rely on ECM channels, in which the photonic flow can travel in a way that is very similar to the one experienced in classic acupuncture [39]. That is why we decided to select specific application protocols involving specific meridians. Links between acupuncture [40], its meridians [37], the extracellular soft tissue matrix and fascia, along with the mechanoreceptors [41], hydraulic transduction [42] and signalling pathways, appear to be quite tight, although robust evidence remains lacking.

## 7. Libralux and Acupuncture

As stated, Libralux stimulus is applied on sequences of acupuncture points. Such points, among the 361 known points, have been selected according to different criteria [43,44,45] for their known effectiveness and for their marginal side-effects because the device has an intended use by Western health operators as well.

The selected APs are described in detail in Table 1.

The above points were used also in the Biolite^®^ application protocols (see reference [7]).

Although the effect of Libralux stimulation strongly resembles the one obtained from classic acupuncture, we cannot declare that the stimulus is equal to the one that can be induced through a classic acupuncture needle. We have, however, found evidence of some successful acupuncture-like treatments [47,48]. Therefore, strictly for acupuncture specialists, a selectable Acupuncture Mode, through which one may select meridian-specific stimulation trains, has been added. By selecting Acupuncture Mode, the 14 meridians are individually selectable (to apply the appropriate meridian modulation), and the operator may apply the stimulus on the desired acupoint. The basic train lasts 20 s and can be repeated or discontinued according to operator choice (See Figure 5).

## 8. Conclusions

In addition to its possible application according to acupuncture criteria, LibraLux could be proposed in a wide range of applications in the areas of musculoskeletal rehabilitation and pain clinics.

Laser therapy contraindications are mainly dependent on power/energy levels, which in the case of Libralux are not applicable. Some contraindications to pulsed stimulations might precautionarily apply to subjects with a previous history of epilepsy and to patients with any type of pacemaker. Further standard restrictions might also apply to oncologic conditions and to pregnancy.

Given the scarce contraindications, Libralux treatment could be proposed for a wide range of patients. Up to 50% of the population faces common musculoskeletal pain, with significant social burden and impacts on activities.

However, it should be noted that no randomized controlled trials are available between Low Level Laser Therapy and Acupuncture, or between Low Level Laser Therapy and Laser Acupuncture. The main reasons include the difficulty of proposing a sham acupuncture or a sham laser acupuncture with visible light radiation. However, the retrospective studies and the animal model tests that are cited in the present paper seem to converge in the indication of a clinical effectiveness still to be appropriately demonstrated.

## 9. Patents

Libralux emission has been patented (It. Patent 102015000073384 dated 7 May 2018).

## Figures and Tables

**Figure 1 medicines-06-00040-f001:**
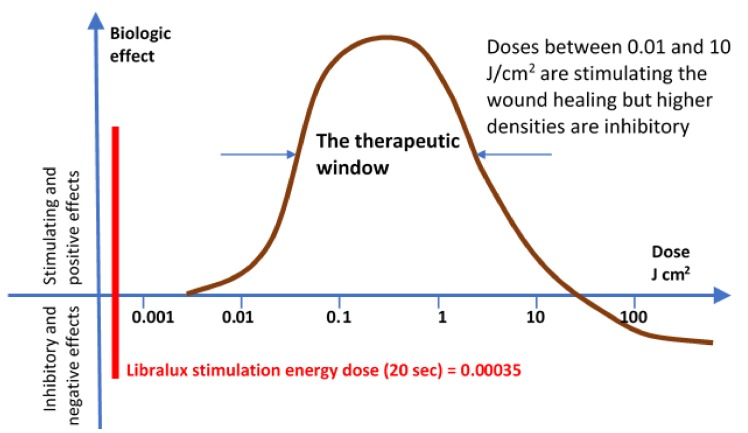
The Arndt and Schulz law for Lasertherapy (modified from https://pocketdentistry.com).

**Figure 2 medicines-06-00040-f002:**
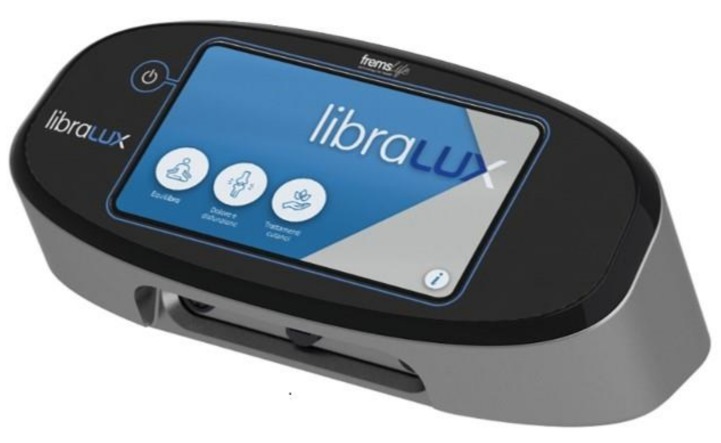
Libralux. (Image by courtesy of Fremslife srl—Genoa, Italy)

**Figure 3 medicines-06-00040-f003:**
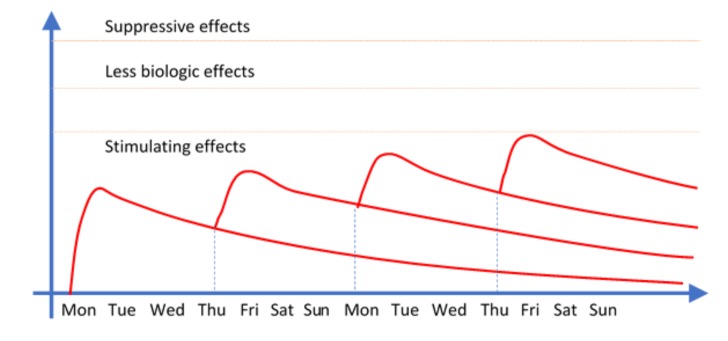
Lasertherapy—biological effects integration (modified from https://pocketdentistry.com).

**Figure 4 medicines-06-00040-f004:**
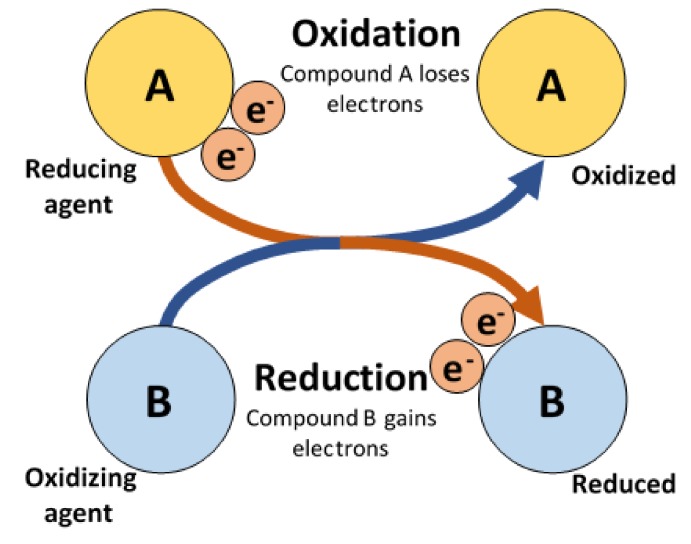
Oxidation and Reduction.

**Figure 5 medicines-06-00040-f005:**
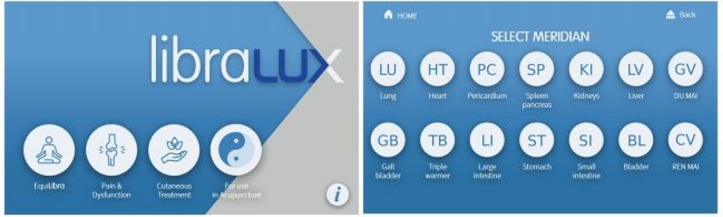
Libralux Acupuncture Mode. Note: (**) (**) Triple Energizer (TE) is still coded as TB (Triple Burner) and Liver LR is still coded as LV with a superseded coding—Software will be amended. (Courtesy by Fremslife Srl, Genoa, Italy)

**Table 1 medicines-06-00040-t001:** Libralux application protocols—Selected Accupoints (*) [46].

CODE	NAME	Anatomic Reference
LI4	**Hegu**	In the middle of the second metacarpal bone on the radial side.
LI5	**Yangxi**	On the radial side of the wrist in a depression between extensor pollicis longus and brevis tendons, found when the thumb is tilted upward.
LI11	**Quchi**	With the elbow flexed, on the radial side of the upper arm at the border of the humerus
TE5	**Waiguan**	2 cun over the dorsal wrist flex crease, between the radius and the ulna.
SI2	**Qiangu**	When a loose fist is made, at the ulnar end of the crease, distal to the fifth metacarpophalangeal joint at the junction of the red and white skin
SI8	**Xiaohai**	Between the olecranon process of the ulna and the medial epicondyle of the humerus, found with the elbow flexed.
LR2	**Xingjian**	On the dorsum of the foot between the first and second toes, proximal to the margin of the web at the junction of the red and white skin.
KI3	**Taixi**	In the depression midway between the tip of the medial malleolus and the attachment of the Achilles tendon
BL60	**Kunlun**	In a depression between the tip of the external malleolus and the Achilles tendon
ST36	**Zusanli**	3 cun below ST35, one finger width lateral from the anterior border of the tibia.
ST38	**Tiaokou**	8 cun below ST35, one finger width lateral from the anterior border of the tibia.
FM23 (**)	**Xiyan**	Lateral and medial knee depressions
BL40	**Weizhong**	Midpoint of the transverse crease of the popliteal fossa, between the tendons of biceps femoris and semitendinosus.
BL66	**Tonggu**	Anterior to the fifth metatarsophalangeal joint.
GB34	**Yanglingquan**	In a depression anterior and inferior to the head of the fibula.

Note: (*) The “cun” is the standard unit of measurement for the body used in acupuncture. As everyone’s body has different dimensions, it is defined according to the person whose body is to be treated or attacked. 1 cun = width of the thumb, in the middle, at the crease. (**) Xiyan is now coded as EM42.

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
