# Peer review of "Ultra-Low-Level Laser Therapy and Acupuncture Libralux: What Is so Special?"

_medicines, 2019, doi:10.3390/medicines6010040_

Round 1
Reviewer 1 Report
This is a great special edition that will add to specific pain conditions. More research needs to be done on laser acupuncture.
Line 47 - What are the different criteria?
Line 88 - when stating Libralux's effectiveness, make sure to not mislead the reader that it is truly effective b/c case studies/pilot studies cannot truly determine that libralux is effective for specific conditions.
Line 91 - Is a gesture similar to a movement? Gesture usually indicates a hand movement or a body movement that expresses some form of meaning/idea or expression. I'm wondering if we can substitute movement or motion instead of gesture as it may fit the topic better.
Line 102-103 : improve wording as "the irrational factor was found and resolved" does not quite make sense as it has both future verb and past tense at the same time.
Section 5.1 - Musculoskeletal pain - may consider adding wording surrounding central sensitization pathology for MSK pain. Define SOL vs. GEL condition.
Section 6 is really excellent addition to the paper.
Conclusion Section: Please make a statement regarding need for future randomized controlled trials/comparative research trials to determine if Libralux is truly effective. There is lacking data for its efficacy with the current case studies/pilot studies.
Author Response
Dear Sir,
also on behalf of the other authors I wish to thank you for your comments and I sincerely hope that our amendements are a satisfactory answer to your suggestions. Hereinafter point-by-point replies.
-----
Line 47 – What are the different criteria
Libralux has been designed on the experiences previously collected with Biolite® radiating a 0.03 mW average power with a 100 Hz, 1% Duty Cycle red light laser emission. The basic idea was to apply a level of energy similar to the one of the solar radiations that did accompany the life evolution. The second one was to pulsate that energy at a frequency in the range of the known physiologic oscillations. The intended effect was an interaction with normal physiology in the extracellular soft tissue matrix rather than producing deep effects on defective tissues.
Line 88 – when stating Libralux's effectiveness, make sure to not mislead the reader that it is truly effective b/c case studies/pilot studies cannot truly determine that libralux is effective for specific conditions.
We submit that the paragraph can hardly be read as a classical clinical evidence as provided by double blind controlled trials. However, the evidences clinical, on animal models and on cultures/cells (see references) are in our opinion quite strong and worth specific consideration. The objective of our review is not to state a specific therapy but rather to underline the strong similarity to acupuncture observed effects achieved by a tiny, pulsed red light laser emission in lieu of needles.
Line 91 – Is a gesture similar to a movement? Gesture usually indicates a hand movement or a body movement that expresses some form of meaning/idea or expression. I'm wondering if we can substitute movement or motion instead of gesture as it may fit the topic better.
Concur. The term gesture, inappropriate, has been modified.
Line 102-103: improve wording as "the irrational factor was found and resolved" does not quite make sense as it has both future verb and past tense at the same time.
Thanks for the suggestion, the phrase has been modified as:
If the pain/dysfunction, in spite of the temporary remission, would return in a few days, the clinician should be aware of an active irritational factor to be found and resolved
Section 5.1 - Musculoskeletal pain - may consider adding wording surrounding central sensitization pathology for MSK pain. Define SOL vs. GEL condition.
The SOL/GEL transition has been very well explained by Pollack (see references – a more specific one has been added) and is describing the physics of the proteins-water colloidal solution filling the extracellular spaces. Both pH and viscosity change from SOL to GEL affecting the cell physiology and perhaps the intrasynaptic exchange.
Section 6 is really excellent addition to the paper.
The approval of the reviewer is especially rewarding!
Conclusion Section: Please make a statement regarding need for future randomized controlled trials/comparative research trials to determine if Libralux is truly effective. There is lacking data for its efficacy with the current case studies/pilot studies.
As suggested a final phrase has been added.
It is however to remark that no randomized controlled trials are available neither between Low Level Laser Therapy and Acupuncture nor between Low Level Laser Therapy and Laser Acupuncture. Main reasons include the difficulty of proposing a sham acupuncture or a sham laser acupuncture with visible light radiation. However, the retrospective studies and the animal model tests that are cited in the present paper seem to converge in the indication of a clinical effectiveness still to be appropriately demonstrated.
Once more thanks for your time
Michele Gallamini
Reviewer 2 Report
The manuscript entitled "ULTRA-LOW-LEVEL LASER THERAPY AND ACUPUNCTURE - LIBRALUX: WHAT IS SO SPECIAL?" reviews the use of very low-intensity laser to treat a musculoskeletal pain by stimulation of distal acupuncture points.
Key words: should be in alphabetical order, KEY WORDS should not contain the same words that are within the title of the text. Thus these should be changed appropriately
This paper is very interesting; however, the conducted review is too overall. The effectiveness of ultra-low laser is strictly connected with the application of the light through acupuncture points. You should compare more deeply the laser acupuncture in contrast to traditional treatment when using needles. Moreover, you should indicate the limitation of the application of LLLT in contrast to LLLT in musculoskeletal pain treatment.
Introduction:
Line 41-43 Add a citation to this statement: “ A certain level of power/energy is required to obtain such effects, but over a given limit, the 41 energy can be destructive or have negative effects, while under a lower limit, it has been 42 demonstrated that no negative effects can be achieved..”
Discussion
Add references comparing ULLLT with traditional acupuncture and LLLT.
Author Response
Dear Sir,
also on behalf of the other authors I wish to express my thanks for your comments. An updated revision is being uploaded.
Hereinafter our point-by-point reply
-------------
Key words: should be in alphabetical order, KEY WORDS should not contain the same words that are within the title of the text. These should be changed appropriately
Keywords have been modified as follows
Laserpuncture; Musculoskeletal disorder; Musculoskeletal pain; Photobiomodulation; Pulse-modulated laser emission.
This paper is very interesting; however, the conducted review is too overall. The effectiveness of ultra-low laser is strictly connected with the application of the light through acupuncture points. You should compare more deeply the laser acupuncture in contrast to traditional treatment when using needles. Moreover, you should indicate the limitation of the application of LLLT in contrast to LLLT in musculoskeletal pain treatment.
Our objective was to tell the 25 years development history of a device moving somehow along an unexplored path. Our efforts have been aimed to support our findings and experiences with a rational model.
Hard to compare a device that by the standing scientific literature lies under any known effectiveness threshold (Tunèr & Hode and all the WALT indications) by several orders of magnitude in terms of power/energy and dosimetry.
We quite quickly found that the stimulus did alter the soft tissue consistency and “thickness” suggesting a likely interaction with the soft tissue extracellular matrix that according to Pollack (see references) in its physiology changes its physical structure and pH.
The application of the stimulus through acupuncture points – an entry point to the ubiquitous extracellular soft tissue matrix – and the observation of the specific paths of diffusion of the stimulus that were resembling the acupuncture meridians guided us in the exploration of acupuncture.
An intermediate paper on Laser in Medical Science (see references) was published and the most recent achievements in ophthalmologic acupuncture as well as on the motor control and balance disorders are on one side requiring the execution of classical double blind controlled trials on specific conditions and our findings may help to better understand the acupuncture and meridian physiology.
Line 41-43 Add a citation to this statement: “ A certain level of power/energy is required to obtain such effects, but over a given limit, the energy can be destructive or have negative effects, while under a lower limit, it has been demonstrated that no negative effects can be achieved..”
A more precise citation has been added and the phrase has been modified as follows.
A certain level of power/energy is required to obtain such effects, but over a given limit, the energy can be destructive or have negative effects, while under a lower limit, it has been demonstrated that no effects can be achieved. The Arndt&Schultz Law [2] stating the principle is shown in Figure 1.
Typical power/energy doses are recommended for these applications by the World Laser Association [3].
Add references comparing ULLLT with traditional acupuncture and LLLT.
We could not find any publication of the kind on PubMed, however, we added the following sentence
It is however to remark that no randomized controlled trials are available neither between Low Level Laser Therapy and Acupuncture nor between Low Level Laser Therapy and Laser Acupuncture. Main reasons include the difficulty of proposing a sham acupuncture or a sham laser acupuncture with visible light radiation. However, the retrospective studies and the animal model tests that are cited in the present paper seem to converge in the indication of a clinical effectiveness still to be appropriately demonstrated.
--------------
Once more thanks for your time
with best regards
Michele Gallamini
Round 2
Reviewer 2 Report
Thank you for the corrections and deep explanation of your work.